# Black Silicon as Anti-Reflective Structure for Infrared Imaging Applications

**DOI:** 10.3390/nano14010020

**Published:** 2023-12-20

**Authors:** Eivind Bardalen, Angelos Bouchouri, Muhammad Nadeem Akram, Hoang-Vu Nguyen

**Affiliations:** Department of Microsystems, University of South-Eastern Norway, Raveien 205, 3184 Borre, Norway; angelos.bouchouri@usn.no (A.B.); muhammad.n.akram@usn.no (M.N.A.)

**Keywords:** black silicon, anti-reflection, infrared imaging, reactive ion etching

## Abstract

For uncooled infrared cameras based on microbolometers, silicon caps are often utilized to maintain a vacuum inside the packaged bolometer array. To reduce Fresnel reflection losses, anti-reflection coatings are typically applied on both sides of the silicon caps.This work investigates whether black silicon may be used as an alternative to conventional anti-reflective coatings. Reactive ion etching was used to etch the black silicon layer and deep cavities in silicon. The effects of the processed surfaces on optical transmission and image quality were investigated in detail by Fourier transform infrared spectroscopy and with modulated transfer function measurements. The results show that the etched surfaces enable similar transmission to the state-of-the-artanti-reflection coatings in the 8–12 µm range and possibly obtain wider bandwidth transmission up to 24 µm. No degradation in image quality was found when using the processed wafers as windows. These results show that black silicon can be used as an effective anti-reflection layer on silicon caps used in the vacuum packaging of microbolometer arrays.

## 1. Introduction

Recently, low-cost thermal cameras targeting the long-wave infrared (LWIR) range (8–14 µm), based on uncooled microbolometer arrays (MBAs), have become widely available [1,2]. The MBA is typically fabricated on a silicon (Si) CMOS readout circuit (ROIC). Since the MBA requires a low-pressure environment (typically below $10^{-2}\;\mathrm{mbar}$) to operate, the MBA on a CMOS chip is typically covered by a Si cap with a hermetic seal. As miniscule leaks and outgassing inside the cavity are inevitable, enlarging the cavity by micromachining the cap will increase the lifetime of the device. Undoped, high-resistive Si is a commonly used material for the cap, having the benefit of being mechanically matched to the ROIC, as well as having relatively low absorption in the LWIR. However, due to the large refractive index of around 3.4 in the IR range, reflection losses at air/vacuum-to-Si interfaces are substantial and therefore require anti-reflection (AR) treatment on the surfaces to increase the signal flux on the sensor. Figure 1 shows a simplified conceptual sketch of a capped MBA-on-CMOS chip.

Conventionally, anti-reflective coatings (ARCs) consisting of multiple thin, optically transparent layers are employed. However, anti-reflective structures based on dense micro- or nanostructures etched in Si are being explored. Examples include anti-reflective gratings (ARGs) [3,4] and moth-eye-like structures [5,6,7,8]. ARGs employ periodic structures such as Si pillars, columns, or cones to act as a metamaterial with an effective refractive index that produce destructive interference, similar to ARCs. Moth-eye structures, on the other hand, achieve reduced reflectance by structuring the surface to create a gradual transition in the refractive index between the air/vacuum and Si.

Black silicon (B-Si) gets its name from its black appearance and is used to refer to silicon with a densely nanostructured surface composed of silicon needles exhibiting moth-eye-like anti-reflective properties. It was originally discovered as an unwanted byproduct of reactive ion etching (RIE). The self-masking effect leading to its formation is primarily attributed to particles unintentionally deposited on a silicon surface or remaining residues due to incomplete removal of a passivation layer [9]. Various other techniques for black silicon formation, including laser irradiation, chemical, wet, or electrochemical etching, can also be employed, resulting in different morphologies and characteristics [10]. Today, researchers have extensively explored the applications of B-Si structuring, with a particular focus on the visible, UV, and near IR ranges. Applications include enhancing the light-collection efficiency of Si photodetectors [11] and solar cells [12,13].

Deep cavities with depths on the order of hundreds of micrometers, crucial for Si caps in MBA vacuum packaging, are routinely created using deep reactive ion etching (DRIE), often employing a Bosch process [14]. This technique involves alternating passivation and etching steps and may produce Si structures with near-vertical sidewalls, for example in micro-electromechanical (MEMS) devices. Since B-Si can be generated in the same DRIE machines, the possibility of processing B-Si on the cap’s backside immediately after cavity formation holds the potential for significant time and cost savings compared to using an ARC or ARG as the anti-reflective structure. An additional advantage is that B-Si can tolerate the high-temperature processing required during the hermetic sealing fabrication step.

Recently, various research groups have examined the properties of B-Si in the mid- to far-infrared ranges (>3 µm). These include investigations on reflectance and the angle dependence of reflectance [15] and potential of using highly doped Si as absorbers [16], achieving near-perfect (99.5%) absorptance at certain wavelengths.

An investigation of B-Si structuring for anti-reflection in the LWIR range was conducted in [5,6,7,8]. In [5,6], B-Si structured by reactive ion etching (RIE) using SF_6_ and O_2_ precursor gases was examined. Various parameters such as pressure, etch time, plasma power, and gas flow ratios were explored, and samples were characterized in terms of transmission through the wafers. Both groups demonstrated the potential to achieve high transmission, approximately 80–90%, in the 8–12 µm range. In [6], it was shown that increasing etch time improves transmission at longer wavelengths while reducing it at shorter wavelengths. This phenomenon is attributed to the deeper etched pores while increasing the pore spacing with prolonged etch time, necessitating a trade-off for the target wavelengths.

Alternatively, ordered moth-eye structures can be fabricated through methods like masking with photoresist patterning [7] or colloidal lithography [8]. However, this approach increases the complexity of fabrication and may not be compatible with processing in deep cavities.

In this work, we intend to expand the investigation of B-Si as anti-reflective layers and to assess their suitability for the purpose of replacing ARCs on Si caps. First, we present a novel etching recipe, based on a gradual pressure reduction during the etching process, resulting in efficient anti-reflective properties across a wide bandwidth. Secondly, we demonstrate that B-Si can be processed in deep cavities, which to the authors’ knowledge has not been previously demonstrated. Thirdly, we explore the effect the impact of B-Si on image quality by measuring the modulated transfer function (MTF) through processed wafers, which to the authors’ knowledge, has also not been demonstrated before.

## 2. Materials and Methods

### 2.1. Materials and Optical Characterization

Double-sided polished (DSP) Czochralski (CZ) Si (100) wafers with a 4-inch diameter, a thickness of 0.52 mm, and a resistivity ranging from 1 to 20 ohm-cm were utilized. Although float zone (FZ) wafers exhibit lower infrared (IR) absorption due to their reduced oxygen content, their cost is prohibitively high for the scope of this development.

The transmittance through the wafers was measured using a Thermo Fisher Scientific (Waltham, MA, USA) Nicolet iS50 Fourier transmission infrared spectrometer (FTIR). Since the B-Si surface is optically rough, incoming waves may scatter at short wavelengths, as the wavelength becomes comparable to the typical pore spacing. However, at longer wavelengths, scattering is negligible. Therefore, for a good approximation, the total transmittance *T* through the wafer at a normal incidence angle, considering multiple internal reflections and losses due to internal absorption, can be described by the following wavelength-dependent equation:(1)$$T\left(T_{1},T_{2},r\right)=T_{1}\cdot T_{2}\cdot r\cdot \sum_{n=0}^{\infty }{\left(1-T_{1}\right)}^{n}\cdot {\left(1-T_{2}\right)}^{n}\cdot r^{2n}=\frac{T_{1}\cdot T_{2}\cdot r}{1-\left(1-T_{1}\right)\cdot \left(1-T_{2}\right)\cdot r^{2}}$$
where $T_{1}$ and $T_{2}$ denote the transmittance at the air-to-Si and Si-to-air interfaces, respectively, while *r* is the relative beam intensity after travelling through the wafer with a thickness *d*, having extinction coefficient κ, for a given vacuum wavelength $\lambda_{0}$:(2)$$r=e^{\left(-4\cdot \pi \kappa d/\lambda_{0}\right)}$$

As the real part of the refractive index (n=η+iκ) for high-resistive Si is well-known in the LWIR range, and the imaginary part of the refractive index is insignificant, the transmittance at the air-Si/Si-air interface can be calculated with Fresnel’s equation to a good approximation using η:(3)$$T_{Si}=1-|\frac{1-\eta_{Si}}{1+\eta_{Si}}{|}^{2}$$

The extinction coefficient κ of the Si wafers used in this work must be determined, as it depends on the doping and impurity levels, which vary widely between wafer types. By measuring the transmittance through four unprocessed wafers, κ was determined using Equations (Equation 1)–(Equation 3). The result for κ as well as the data used for η are shown in Figure 2. For the four wafers measured, a significant variation in absorption was observed, as indicated by the error bars. The average value for κ was used in all following calculations. The absorption bands around 9 µm and 19 µm in κ are known to be due to oxygen complexes in Si [17,18], while the remaining bands above 7 µm, including large peak near 16 µm, can be ascribed to lattice absorption [19]. It can be seen that the measured absorption peaks around 9 µm is higher than the referenced result, while the absorption peaks around 16 µm is lower. The former may due to the higher oxygen content in CZ wafers.

B-Si structures were inspected and characterized by means of a scanning electron microscope (SEM, Hitachi High-Tech Corporation (Ibaraki, Japan) SU3500).

### 2.2. Sample Processing

In order to approach the conditions used in a realistic process, the following steps were performed when processing the wafers:Step 1: Masking the wafers with photoresist. AZ4562 photoresist with a thickness of 15 µm was spun on the wafers. The mask consisted of square openings with widths ranging from 11 mm to 15 mm. The relatively thick photoresist was necessary due to the high erosion of photoresist during the cavity and B-Si etch.Step 2: Si etching. A DRIE process was used to etch square openings into the cavities. For unmasked wafers, the entire wafer surface was etched.Step 3: Etching of B-Si. The B-Si was etched into the cavities directly or shortly after the cavity etch.Step 4: Oxide etch, performed directly, or shortly after the B-Si etch.

For some samples, where noted, the wafers were left unpatterned, resulting in a B-Si surface covering the wafers. For these wafers, step 1 was omitted, and etching was performed on the original wafers. Etching steps 2–4 were performed using an Oxford Instruments (Bristol, United Kingdom) Estralas DRIE machine with inductively coupled plasma (ICP) source. The temperature of the wafer was controlled by backside helium cooling, set at 0 °C for all etching steps. For the Si etch step (step 2), a Bosch process with SF_6_ and C_4_F_8_ precursor gases was used for the etching and passivation, respectively. Samples with both shallow (<20 µm) and deep cavities (>100 µm) were fabricated. For B-Si etching (step 3), a mixed process with SF_6_ and O_2_ precursor gases was used. During the initial experiments, it was found that a gradual reduction in pressure over time resulted in higher pore depths for the same density, compared to a process with constant pressure. This led to a nominal B-Si recipe of 11 min duration where the pressure was reduced gradually from 35 mTorr to 25 mTorr (1 mTorr = 1.33 mbar) in three substeps. Additionally, after the B-Si etch, an oxide etch (step 4) was used to clear the $SiF_{x}O_{y}$ passivation layer remaining after the B-Si etch.

Table 1 details the etch parameters for steps 3 and 4. For steps 3-1 to 3-3, the effect of total etch time was investigated by scaling the etch time by a factor *x*, while keeping all other parameters fixed. Similarly, the oxide etch time $t_{4}$ was varied in the experiments.

Table 2 shows the etch parameters for the samples referenced in Section 3. All samples were processed as wafers, except #8 and #9, which were pre-diced into 3 cm × 3 cm chips. These samples were coated with an ARC on one side and two sides, respectively, fabricated previously [21]. On the one-sided ARC sample, B-Si was processed on the other side by placing it on a bare Si wafer with a thermally conducting oil between during etching.

### 2.3. Characterization of Imaging Performance

The optical performance of single sided B-Si wafers was evaluated using the slanted edge method to extract the modulation transfer function (MTF). The MTF was calculated using the freely available code “sfrmat5” [22] and has been used to measure the optical performance of anti-reflective metasurfaces in the past [4]. In the slanted edge method, a razor-sharp edge is placed at a small angle in relation to the camera’s pixel grid. The edge is positioned in front of an object that has high contrast relative to the edge, such as black and white for the visible spectrum or ‘hot’ and ‘cold’ for thermal cameras and in focus. Both hot and cold areas on the target must be uniform for this method to give reliable and correct results. From the tilted edge image, it extracts the oversampled edge intensity profile, after which Fourier transform is performed. The result, the MTF, is a basic characteristic property and describes the “sharpness” of an optical system. An extensive and descriptive analysis of the slanted edge method can be found in [23], where the MTF is calculated on various IR cameras and under different conditions. Surfaces such as a patterned single-sided B-Si can be seen as thin random phase screens, and their optical effect can be modeled as an increase in the wavefront error and a reduction in the MTF [24].

The uncooled LWIR thermal camera PLUG612R with a pixel pitch of 12 μm from Simtrum was used, with a lens with fixed focal length f=19 mm and F/number = 1.0. Its total angular field of view (AFOV) is 22.9${}^{{}^{\circ}}$, which was calculated from $AFOV=tan^{-1}\left(W/\left(2*f\right)\right)$, where *W* is the sensor width (pixel count · pixel pitch). The “QuickCal” black-body from ISOTECH, which has a circular target size of 25 mm, was used. It was placed 62 cm in front of the camera and kept at 100 °C. A sharp razor edge was placed 3 mm in front of the black-body, at approximately 6${}^{{}^{\circ}}$ relative to the pixel grid, and the camera was focused on it. The captured images were saved in “grayscale” 8-bit uncompressed TIFF format. All software enhancements were disabled except the time domain filter. The camera was initially calibrated using the black body without any sample between the black body and the camera. The camera’s color range was set from 0 °C for black to 115 °C for white.

The camera has a sampling frequency of 83.3 cycles/mm and a corresponding Nyquist frequency of 41.65 cycles/mm. The Nyquist frequency is the limit up to which the optical information can be correctly retrieved. Optical information above the Nyquist frequency is aliased down to lower frequencies. Figure 3 illustrates the setup for the optical measurements. We examined the optical performance of one-sided B-Si wafers in two situations, where in both cases, the B-Si surface was facing the camera.

In Case A, the B-Si wafer was placed approximately at the object plane, 3 mm in front of the razor edge. This case simulates the situation when the wafer is used as an encapsulation cap and very close to the image plane in the packaged bolometer sensor. In this case, the incoming light will propagate through the cap under different angles, with the largest angle determined by the F/number of the lens. This case is more challenging than Case B since the rough B-Si surface is in close proximity to the image plane and scattering can reduce the MTF values significantly.

In Case B, the B-Si wafer was placed near the entrance pupil plane at a distance of 10 mm in front of the camera. This simulates the situation where the wafer acts as a transmission window and can be used as outer protection in an optical system. In this situation, the object plane is usually located far away and the incoming light is almost parallel, thus the scattering surface is out of focus.

Both on-axis and off-axis measurements were taken in both cases. The on-axis measurements were taken with the optical axis towards the black-body source. Off-axis measurements were taken by rotating both the camera and wafer by 9${}^{{}^{\circ}}$, keeping the wafer perpendicular to the optical axis.

## 3. Results

### 3.1. IR Transmission through Processed Wafers

#### One-Sided B-Si—Effect of Etch Duration

In the following, the results are presented for one-sided B-Si in shallow cavities (<20 µm depth), where the etch time of steps 3 and 4 were varied. As shown in Figure 4a, for all samples, a sharp drop in transmission towards shorter wavelengths is evident, consistent with loss by scattering. The cut-off wavelength is well below 8 µm for all samples. A clear general trend of higher transmission at long wavelengths, at the expense of higher cutoff wavelengths, is also observed. SEM images for the sample with the highest transmission at the longest wavelengths (sample #2) is shown in Figure 4b,c, where pore depths up to 5 µm are seen in Si.

The effect of the oxide etch (step 4) is demonstrated in Figure 5. For the sample processed without the oxide etch step, there are prominent dips in transmission centered around approximately 8.1 µm and 9.1 µm. While the 9.1 µm valley is likely due to the oxygen content, the cause of the additional valley at the lower wavelength is unclear, although it may be related to the fluoride content in the $SiF_{x}O_{y}$ layer left after the B-Si etch. After the final oxide etch step, the absorption is significantly reduced, similar to that of the unprocessed wafer. The effect of the oxide etch also remained after long-time storage in air, with only a slight lowering of the transmission around the 9.1 µm dip observed after 7 months. The slightly lower overall transmission for the stored sample may be due to measuring at different areas on the wafer which has non-uniformities in the B-Si morphology after etching. It can also be seen that the relatively long 4 min oxide etch step increases the transmission of long wavelengths while shifting the cut-off to a longer wavelength, indicating that the oxide etch further deepens the pores while increasing the characteristic spacing between pores.

### 3.2. Double-Sided B-Si and Comparison with ARC

Based on the results described above, a wafer with double-sided (DS) B-Si was prepared (sample #7 in Table 2). In addition, a sample with ARC on one side and B-Si on the other was prepared (sample #8). The transmission spectra for both samples and a DS ARC sample (sample #9) were measured at incidence angles $0^{{}^{\circ}}$, $15^{{}^{\circ}}$ and $30^{{}^{\circ}}$.

As shown in Figure 6, the DS B-Si wafer shows high transmission from about 7 µm to 24 µm. The drop in transmission when increasing the incidence angle up to $30^{{}^{\circ}}$ is less than 2% up to 13 µm and increasing at longer wavelengths. In comparison, the DS ARC sample shows higher angle-dependence, although the change is less that 2% in the 8–12 µm range, while the ARC-BSi sample shows an intermediate behaviour.

### 3.3. B-Si in Cavities

Two samples with B-Si in deep cavities with 390 µm (sample #10) and 180 µm (sample #11) depth, respectively, were fabricated. The measured transmission is shown in Figure 7a. Although the two samples were similar in terms of overall transmission, the thinner Si diaphragm of sample #10 led to less internal absorption for the characteristic 9 µm and 16 µm valleys. The reflectance (R=1−T) of the B-Si surfaces in cavities and the results for corresponding samples in shallow cavities were calculated by solving Equation (Equation 1). The results are an approximation as the extinction coefficient of the processed wafers is likely affected by the etch process, while they are invalid at the lower wavelengths due to scattering. The results are shown in Figure 7b, where it can be seen that the reflectance is lower than 5% in the 8–12 µm range for sample #5. For this recipe, the sample with 180 µm cavity yielded a similar result, with a deviance lower than 3 percentage points between wavelengths from 6 µm to 22 µm. In comparison, for the 390 µm deep cavities, there is a significant difference, with a lower cut-off and higher reflection at longer wavelengths than for the corresponding recipe for shallow cavity. This indicates a deviance in B-Si structure due to the deeper etch, although more experiments would be needed to evaluate if this is a systematic effect.

The cross-sectional SEM images for the B-Si in the 390 µm deep cavity are shown in Figure 8a. Consistent with the transmission results, the pores are less deep than those for the corresponding recipe in shallow cavity, as seen in Figure 4b. Near the sidewall, the direction of the B-Si pores is angled towards the center. As shown in Figure 8b, the cavity edge profile is curved, with a negative wall angle. Thus, the cavity recipe should be optimized in a future process to obtain a plane surface to avoid beam disturbance. However, this was not the focus of this work.

### 3.4. Imaging Performance

Samples #1 and #5, which had unpatterned B-Si on one side, were used in the optical measurements. For the purpose of comparison, measurements were also taken with no wafer and an unprocessed DSP wafer.

#### 3.4.1. Case A (Wafer at Object Plane)

The measured MTF for Case A is shown in Figure 9a. For the on-axis, all cases were performed identically. There was no significant difference between no wafers, DSP wafers, and the single-sided B-Si wafers in terms of the measured MTF.

For the off-axis measurement, the MTF for the no-wafer, DSP and B-Si samples performed equally again, with no significant difference observed between the samples. There is a small degradation in the MTF, which is expected as the optical performance of lenses tends to degrade for off-axis imaging. The slight off-axis degradation was observed across all samples, including the “no wafer” measurement.

These measurements indicate that the rough B-Si wafer does not cause any significant scattering of the incoming wave at the measured wavelengths, and hence, its effects on the image quality are negligible.

#### 3.4.2. Case B (Wafer at Pupil Plane)

The measured MTF for Case B is shown in Figure 9b. For the on-axis, the B-Si was performed with no significant difference in terms of the measured MTF below the Nyquist frequency when compared to no wafers and DSP wafers. Above the Nyquist frequency, the performance started to differentiate. For the off-axis result, the MTF values for B-Si are not degraded as compared to the no-wafer case. Again, these measurements show that the B-Si wafer does not create any scattering of light into higher diffraction orders at the measured wavelength; thus, the image quality is preserved.

## 4. Discussion

In Figure 10, the result for DS B-Si produced in this work is compared to other recently published results for ARCs and ARGs, as well as B-Si. In the 8–12 µm range, our DS B-Si samples exhibited performance on par with state-of-the-art ARCs and ARGs recently demonstrated. Compared with the two previously published results for B-Si, our results are comparable or better, although due to the use of different wafer types, an exact comparison of performance cannot be made.

In terms of total transmission, the results show that B-Si structuring gives comparable results to the state-of-the-art ARCs and ARGs in the 8–12 µm wavelength range. A potential advantage of B-Si, however, is the possibility to obtain much broader bandwidths. Our results demonstrated high transmission and low corresponding reflectance down to 10% up to 24 µm wavelength. Although wavelengths above 14 µm are not typically targeted for thermal cameras due to the high absorption in the atmosphere of these wavelengths, a Si window transparent at these wavelengths could be of use for certain niche applications, such as monitoring of the Earth’s atmosphere in the far infrared (>15 µm) [26].

A more interesting route could be to optimize the B-Si for the 3–5 µm midwave infrared (MWIR) range. Our results demonstrated high transmission down to 3 µm (e.g., as seen for sample #1 in Figure 4a, compared to the unprocessed Si wafer, although this comes at the expense of reduced transmission for the longer wavelengths. Additionally, scattering, and thereby blurring effects, would be more pronounced, as indicated by the sharp drop in transmission at the lower wavelengths. By further developing and tuning recipes to obtain denser structures with high needle depths, it could become feasible to also cover the MWIR range with B-Si structured caps.

The possibility of replacing ARCs in the vacuum cavity with B-Si requires several additional investigations. A critical factor is the need to avoid fracturing of the mechanically fragile Si needles, as such loose needles might damage the bolometer array. Additionally, contact with dust particles must be avoided, as they are not easily removed by blowing or washing. Thus, the use of B-Si as an anti-reflective layer requires it to be sealed off from dusty environments.

The B-Si process must be compatible with the other steps in the cap fabrication and assembly. These steps include metal and getter deposition and cleaning steps. Furthermore, the outgassing of the processed surfaces must not be higher than for unprocessed Si or ARC. Potentially, the large surface area of B-Si may have an effect on the outgassing characteristics. The impacts of the MBA vacuum sealing process and the resulting deformation of cap membrane on B-Si performance also needs to be investigated. Additionally, optimization of the cavity etch recipe, including finding the suitable etch depth and etch profile optimization may be required.

The optical characterization indicated that in the 8–12 µm range, the tested B-Si surfaces would have no detrimental effect on the image quality. Although the optical setup was not identical to the real situation, where the B-Si would be located a few hundred micrometer above the detector array, the results, in addition to the measured transmission curves, give confidence that scattering would not affect the image quality.

## 5. Conclusions

B-Si has been demonstrated as a promising anti-reflective structure on Si caps for the vacuum packaging of MBA. A novel etch recipe for B-Si was developed and applied on both sides of Si wafers and in deep cavities of up to 390 µm depth. The anti-reflection performance of the B-Si is competitive to conventional ARCs in the LWIR. However, B-Si could offer a wider bandwidth with good transmission up to 24 µm and down to 3 µm. In the LWIR range, the image quality obtained by an IR camera, which was characterized by means of the modulation transfer function, show no negative effect of the B-Si on image quality and demonstrates its applicability as a suitable anti-reflective structure, although further investigations are needed to validate its applicability.

## Figures and Tables

**Figure 1 nanomaterials-14-00020-f001:**
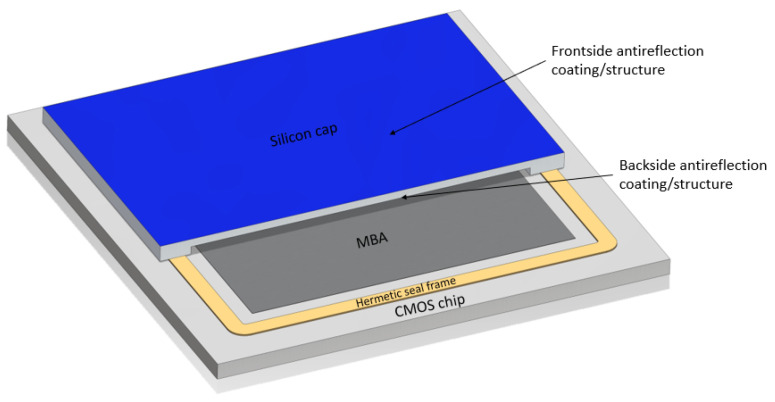
Simplified sketch of a typical MBA-on-CMOS with Si cap for hermetical sealing. The cap is cut to reveal the interior cavity. Dimensions are not to scale. For instance, the width of the MBA sensor area might be approximately 10 mm for a resolution of 640 × 512 pixels and pixel size of 17 µm × 17 µm.

**Figure 2 nanomaterials-14-00020-f002:**
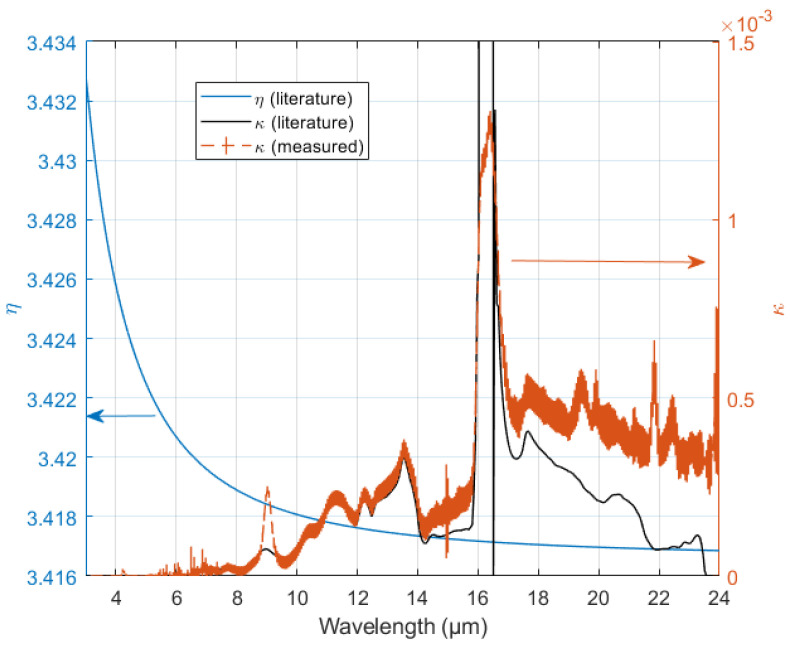
Real part (η) of the refractive index for Si (solid blue line) [20], calculated extinction coefficient from the measurement of four wafers (dashed line with error bars), and extinction coefficient for a high-purity Si wafer (solid orange line) given by [20].

**Figure 3 nanomaterials-14-00020-f003:**
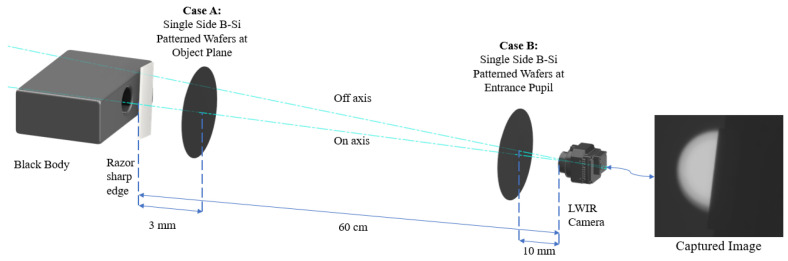
Setup for capturing images using slanted edge method. The wafer is positioned either near the razor edge (Case A) or near the camera lens (Case B). The razor edge is rotated by 6${}^{{}^{\circ}}$ relative to the vertical axis, as shown in the captured image. On-axis measurements are performed with the camera detector array pointing directly to the black-body source, while off-axis measurements are performed by rotating the camera by 9${}^{{}^{\circ}}$.

**Figure 4 nanomaterials-14-00020-f004:**
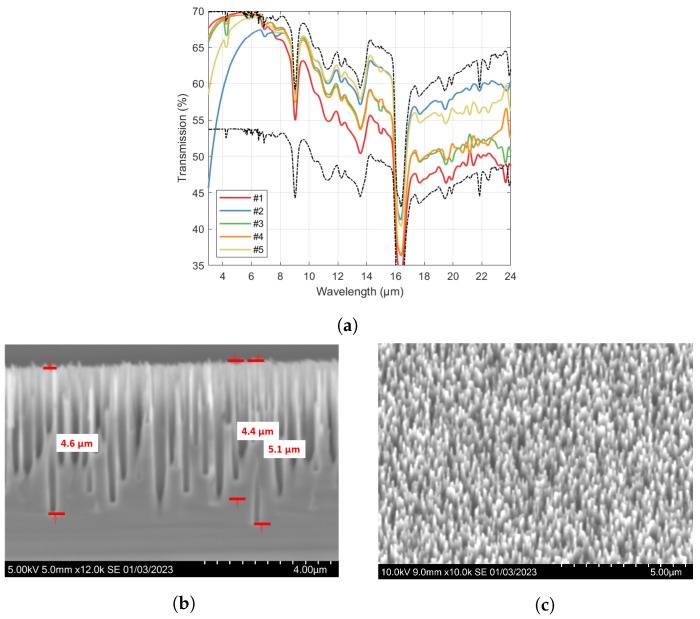
(**a**) Transmission through Si wafers with approximately 500 µm thickness with B-Si on one side. Upper and lower dashed lines show the theoretical transmission through 500 µm thick Si corresponding to perfect ARC (Equation (Equation 1) as $T(1,T_{Si},r)$) on one side and no ARC (Equation (Equation 1) as $T(T_{Si},T_{Si},r)$), respectively. Numbers in the legend refer to the sample number in Table 2. (**b**,**c**) SEM images of sample #2 taken at cross-section (**b**) and at 15${}^{{}^{\circ}}$ (**c**).

**Figure 5 nanomaterials-14-00020-f005:**
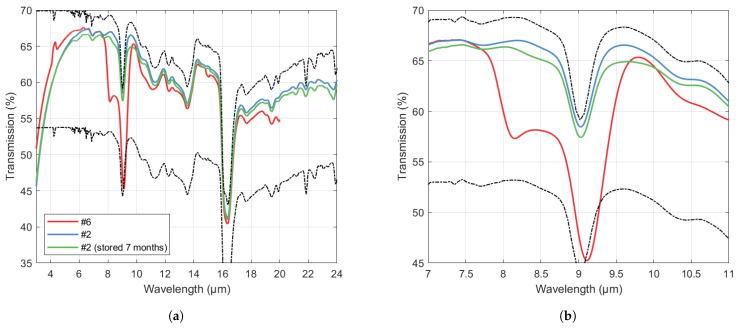
(**a**,**b**) Transmission for one-sided B-Si with no oxide etch (sample #6), 4 min oxide etch (sample #2), and 4 min oxide etch stored 7 months in ambient atmosphere. Upper and lower dashed lines show the theoretical transmission through 500 µm thick Si with perfect ARC on one side and no ARC, respectively. (**b**) shows the zoom-in the wavelength range 7–11 µm.

**Figure 6 nanomaterials-14-00020-f006:**
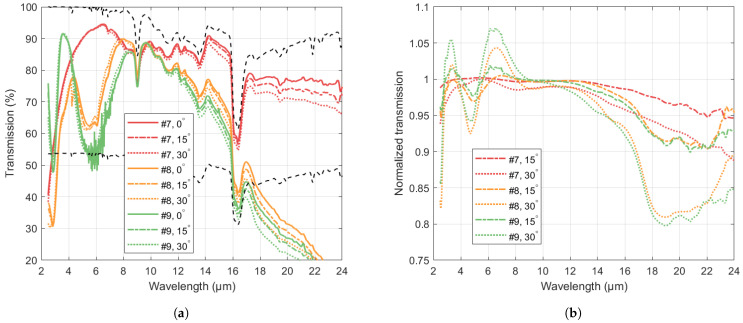
(**a**) Transmission through Si wafers with AR treatment on both sides for incidence angles $0^{{}^{\circ}}$, $15^{{}^{\circ}}$, and $30^{{}^{\circ}}$ (sample #7: DS B-Si, sample #8: ARC on one side, B-Si on the other side, sample #9: DS ARC). Upper and lower dashed lines show the theoretical transmission through 500 µm thick Si with perfect DS ARC (Equation (Equation 1) as T(1,1,r)) and no ARC, respectively. (**b**) Transmission for $15^{{}^{\circ}}$ and $30^{{}^{\circ}}$ incidence angle, normalized to transmission for $0^{{}^{\circ}}$.

**Figure 7 nanomaterials-14-00020-f007:**
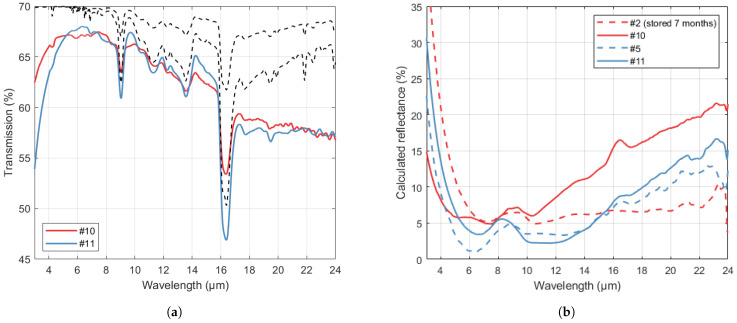
(**a**) Transmission through samples with B-Si in cavities with depth of 390 µm (sample #10) (Si thickness 130 µm) and 180 µm (sample #11) (Si thickness 340 µm), respectively. The upper and lower dashed lines represent the transmission through Si with thicknesses of 130 µm and 340 µm, assuming theoretically perfect (one-sided) AR. (**b**) Calculated reflectance (R=1−T) for the samples with deep cavities (samples #10 and #11, solid lines) and results for corresponding recipes with shallow cavities (dashed lines). Note: The results in (**b**) were smoothed with a moving average filter (period of 1 µm).

**Figure 8 nanomaterials-14-00020-f008:**
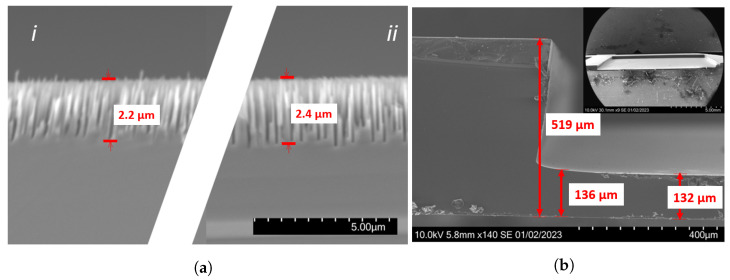
(**a**) SEM images of B-Si sample #10 near cavity wall (**i**) and near center (**ii**) of cavity for sample with 390 µm deep cavity (sample #10). Scale bar is common for (**i**,**ii**) in (**a**). (**b**): SEM image showing overview of cavity edge and full cavity (inset).

**Figure 9 nanomaterials-14-00020-f009:**
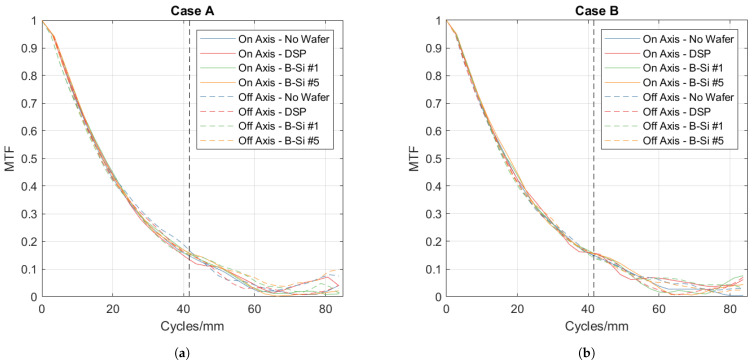
The measured MTF performance for (**a**) Case A and (**b**) Case B. For (**a**,**b**), the dashed black line represents the Nyquist frequency.

**Figure 10 nanomaterials-14-00020-f010:**
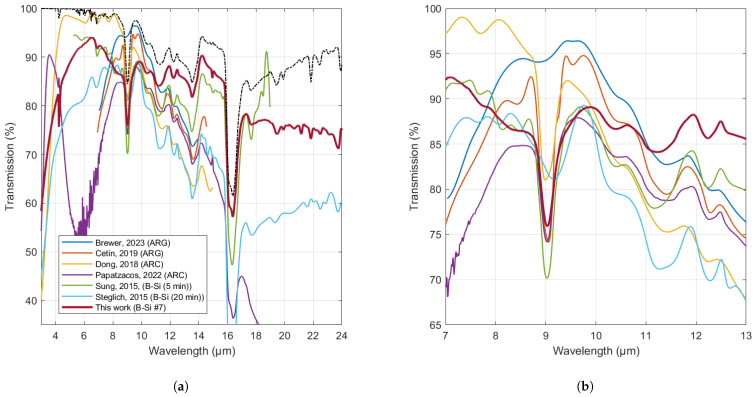
(**a**,**b**) Comparison between results obtained in this work and previously published results * for (double-sided) ARCs, ARGs, and B-Si. Dashed line shows the theoretical maximum transmission for perfect (double-sided) ARC for the wafers used here. (**b**) shows the zoom-in of graph (**a**) in the range 7–13 µm. Note: Referenced works use CZ Si wafers with 0.5 mm thickness, except (Brewer, 2023) and (Steglich, 2015), who used float zone wafers with 0.5 mm thickness and (Sung, 2015), who used CZ wafers with thickness 345–400 µm. Referenced data were extracted as data points from the published graphs using an online tool (https://apps.automeris.io/wpd (accessed on 13 September 2023)) and interpolated. * Referenced results can be found in [3,4,5,6,21,25].

**Table 1 nanomaterials-14-00020-t001:** Etch parameters used for the B-Si (step 3-1 to 3-3) and oxide etch (step 4). *x* denotes the time scaling of steps 3 and 4, respectively. For example, for *x* = 1.5, the etch time is 4.5, 6, and 6 min for steps 3-1, 3-2, and 3-3 respectively, giving a total etch duration of 16.5 min for step 3. $t_{4}$ denotes the etch duration for step 4.

		B-Si Etch		Oxide Etch
Parameter	Step 3-1	Step 3-2	Step 3-3	Step 4
ICP Power (W)	1500	1500	1500	1500
Plate power (W)	50	50	50	50
SF_6_ gas flow (sccm)	36	36	36	0
O_2_ gas flow (sccm)	47	47	47	0
Ar gas flow (sccm)	0	0	0	36
CHF_3_ gas flow (sccm)	0	0	0	17
Pressure (mTorr)	35	30	25	10
Time (min)	3 · *x*	4 · *x*	4 · *x*	$t_{4}$

**Table 2 nanomaterials-14-00020-t002:** Parameters used for processed samples.

Sample Number	Processing (Frontside, Backside)	B-Si Etch Time (min:s)	Oxide Etch Time t_4_ (min:s)	Si Thickness after Etching (µm)
#1	(B-Si ${}^{†}$, Bare)	8:15	1:00	500
#2	(B-Si, Bare)	11:00	4:00	500
#3	(B-Si ${}^{†}$, Bare)	11:00	1:00	500
#4	(B-Si ${}^{†}$, Bare)	13:45	1:00	500
#5	(B-Si ${}^{†}$, Bare)	16:30	1:00	500
#6	(B-Si, Bare)	11:00	0:00	500
#7	(B-Si, B-Si)	13:45 *	1:30 *	480
#8	(B-Si, ARC)	13:45	1:30	500
#9	(ARC, ARC)	N/A	N/A	520
#10	(B-Si in deep cavity, Bare)	11:00	4:00	130
#11	(B-Si in deep cavity, Bare)	16:30	1:00	340

${}^{†}$ Unpatterned. * Processed on both sides.

## Data Availability

Data sets for FTIR measurements available at https://doi.org/10.23642/usn.24865689.v1 19 December 2023. (DOI: 10.23642/usn.24865689).

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
