# Peer review of "Black Silicon as Anti-Reflective Structure for Infrared Imaging Applications"

_nanomaterials, 2023, doi:10.3390/nano14010020_

Round 1

Reviewer 1 Report

Comments and Suggestions for Authors

The authors submitted "lack silicon as anti-reflective structure for infrared imaging applications" Here are my comments:

1. The schematic cartoon to show the idea of this work should be provided.

2. The novelty of the work should be presented in the introduction part.

3. The quality of figures are not good enough and the reviewer can not review them.

4. References part is too few and this is important part to show the novelty of the work.

Comments on the Quality of English Language

 Moderate editing of English language required

Author Response

  1. schematic cartoon was added
  2. the introduction has been rewritten 
  3. It is not clear what improvements can be made to the figure, as no specific advice has been given. 
  4. Several references has been added to the introduction

Besides, extensive proof-reading was done to fix language.

Regarding the concerns about research design, methods, results, and conclusions, as no specific advice or examples was given, it is difficult for the authors to make changes with respect to these points.  

Reviewer 2 Report

Comments and Suggestions for Authors

See attached file

Author Response

  1. References were added
  2. Explanation has been added
  3. Formula has been expanded
  4. Comment+reference about this has been added
  5. An example with reference was added

Reviewer 3 Report

Comments and Suggestions for Authors

The work was written efficiently and in a very clear way. Addresses important issues related to anti-reflection in cameras operating in the infrared range. I have some minor comments regarding Fig. 1 - maybe we need different colors for the presented curves to make it more readable. Line 118 - he pressure was reduced gradually from 35 mT to 25 mT - what are these units?, this needs to be explained

Author Response

All concerns were addressed:

  • Figure 1 (2 in the new document) was modified.
  • Units were fixed and explained

Round 2

Reviewer 1 Report

Comments and Suggestions for Authors

This manuscript could be accepted in this journal.